# Acoustic Assessment of Multiscale Porous Lime-Cement Mortars

**DOI:** 10.3390/ma16010322

**Published:** 2022-12-29

**Authors:** Irene Palomar, Gonzalo Barluenga

**Affiliations:** Department of Architecture, Universidad de Alcalá, 28801 Madrid, Spain

**Keywords:** lime-cement mortar, polymer fibers, expanded clay, perlite, vermiculite, airborne noise, sound absorption, sound insulation, multiscale porosity model

## Abstract

Noise pollution is an issue of high concern in urban environments and current standards and regulations trend to increase acoustic insulation requirements concerning airborne noise control. The design and development of novel building materials with enhanced acoustic performance is an efficient solution to mitigate this problem. Their application as renders and plasters can improve the acoustic conditions of existing and brand-new buildings. This paper reports the acoustic performance of eleven multiscale porous lime-cement mortars (MP-LCM) with two types of fibers (cellulose and polypropylene), gap-graded sand, and three lightweight aggregates (expanded clay, perlite, and vermiculite). Gap-graded sand was replaced by 25 and 50% of lightweight aggregates. A volume of 1.5% and 3% of cellulose fibers were added. The experimental study involved a physical characterization of properties related to mortar porous microstructure, such as apparent density, open porosity accessible to water, capillarity absorption, and water vapor permeability. Mechanical properties, such as Young’s modulus, compressibility modulus, and Poisson’s ratio were evaluated with ultrasonic pulse transmission tests. Acoustic properties, such as acoustic absorption coefficient and global index of airborne noise transmission, were measured using reduced-scale laboratory tests. The influence of mortar composition and the effects of mass, homogeneity, and stiffness on acoustic properties was assessed. Mortars with lower density, lower vapor permeability, larger open porosity, and higher Young’s and compressibility modulus showed an increase in sound insulation. The incorporation of lightweight aggregates increased sound insulation by up to 38% compared to the gap-graded sand reference mixture. Fibers slightly improved sound insulation, although a small fraction of cellulose fibers can quadruplicate noise absorption. The roughness of the exposed surface also affected sound transmission loss. A semi-quantitative multiscale model for acoustic performance, considering paste thickness, active void size, and connectivity of paste pores as key parameters, was proposed. It was observed that MP-LCM with enhanced sound insulation, slightly reduced sound absorption.

## 1. Introduction

Airborne noise is a significant issue in urban environments that has been associated with urban pollution and health problems [1]. To tackle these problems, new building and urban regulations have upgraded the requirements to reduce noise production and improve acoustic insulation [2]. Most of the existing buildings do not meet those requirements due to their low acoustic performance and must be refurbished in due course to reduce noise and improve urban health.

Noise reduction in buildings refers both to the effect on the urban environment and inside buildings. In the first case, sound absorption of coating materials, which depends on surface continuity and material open porosity connectivity, can reduce sound reflection and therefore urban acoustic pressure [3,4]. Sound transmission from outside to the interior of the buildings depends on sound insulation, which is related to mass, homogeneity, stiffness, and material continuity of the building enclosure [5].

The application of new rendering mortars with improved acoustic properties has been reported to be an effective way to reduce airborne noise in existing buildings because their properties can be regulated by modifying their composition [6]. Mortars and other conglomerated materials have been usually considered good sound insulation materials due to their high density, although bad noise absorbents are due to their high acoustic reflection [7]. However, the acoustic performance of cement-based materials depends on the size and distribution of pores and variation in components’ properties [8,9,10]. Acoustic absorption of mortars can be enhanced by selecting aggregate grading and maximum aggregate size [11]. Including other mortar components, such as foam agents [10], rubber [12,13], cork, expanded clay [14], coke [1], perlite and vermiculite [15,16], carbonized lightweight bio-based aggregates [17], recycled plastic [18], cellulose or polypropylene fibers [10,19,20,21], has also shown to be an effective way to improve acoustic behavior of mortars. To evaluate the acoustic properties of mortar samples, some authors have proposed reduced-scale testing procedures that can be used for comparative purposes [12,14,17,22].

The present study aims to evaluate the acoustic performance of lime-cement mortars regarding airborne noise using reduced-scale tests. The tasks for achieving this goal are:

Discussing the effect of gap-graded aggregates, lightweight aggregates, and fibers on airborne noise absorption and acoustic insulation of porous lime-cement mortars.Considering the influence of physical and mechanical parameters over sound insulation performance of multiscale porous lime-cement mortars.Analyzing the relationship between sound absorption and sound insulation parameters of multiscale porous lime-cement mortars.Describing a multiscale semi-quantitative model for the acoustic performance of porous lime-cement mortars.

## 2. Experimental Program

Eleven lime-cement mortar mixtures were evaluated, studying physical properties related to the material’s porous structure, mechanical properties measured by ultrasonic pulse transmission, and noise absorption and insulation properties.

### 2.1. Materials

The components used for the mortar compositions were:A binder mixture of an aerial lime type CL-90 S, designated according to the European standard UNE-EN 459-1 [23], and a white cement type BL-II/B-L 32,5 N, designated according to the European standard UNE-EN 197-1 [24] and the Spanish standard UNE 80,305 [25].Two types of siliceous sand: a continuous particle size distribution of 0–4 mm (CGA) and a gap-graded particle size distribution of 2–3 mm (GGA), characterized by the lack of particles under 2 mm.Three types of lightweight aggregates (LA): expanded clay (A), perlite (P), and vermiculite (V).Two types of short fibers: cellulose fibers (CF) of 1 mm in length and 20 µm diameter and polypropylene fibers (PPF) of 6 mm in length and 30–35 µm in diameter.

Table 1 summarizes the eleven lime-cement mortar compositions used in this study. The proportion in volume lime to cement to aggregate remained 1:1:6 for all mortars. The amount of water was fixed in each case to achieve a plastic consistency [26]. It can be pointed out that no polymeric plasticizer was used. Two reference mortars with continuous-graded (REF) and gap-graded (REFC) natural siliceous aggregates were designed. Three lightweight aggregates were used to replace 25 and 50% of natural aggregate, resulting in five new mortar compositions (only 25% of replacement was considered for expanded clay). Cellulose fiber and PP fiber were added to the gap-graded reference and the mixture with 25% of perlite, resulting in four new mixtures.

### 2.2. Experimental Methods and Preliminary Results

#### 2.2.1. Physical Properties Characterization

Physical characterization of the porous structure of the mortar samples was performed to measure apparent density (D_AP_), open porosity accessible to water (P_O_), capillary water absorption (C), and water vapor permeability (P_V_), according to the European standards UNE-EN 1015-10, UNE-EN 1015-18, and UNE-EN 1015-19, respectively [27,28,29]. Table 2 presents the physical parameters experimentally measured on mortar samples, which have been previously correlated [6].

#### 2.2.2. Mechanical Characterization

Ultrasonic pulse velocity transmission of 250 kHz compressive (p-) and shear (s-) waves were used to calculate Young (E) and compressibility (K) moduli and Poisson’s ratio (ν), according to the methods described elsewhere [30]. Table 3 records the experimental results obtained to characterize elastic stiffness and compressibility parameters.

#### 2.2.3. Acoustic Characterization

Two parameters were used to characterize the acoustic performance of mortars. Noise reduction coefficient (α_NRC_) was calculated using an impedance tube test (Figure 1) on samples of 96 ± 2 mm diameter and 40 ± 2 mm thickness (UNE-EN ISO 10534-2 [31]) and frequencies ranging from 50 to 1600 Hz [6].

Noise insulation was experimentally evaluated with the sound reduction index (R_A_) measured in a reduced scale acoustic chamber [12,32] on 220 × 240 mm^2^ mortar samples with a thickness of 24 ± 2 mm (Figure 2). The acoustic chamber (Figure 3) consisted of two compartments acoustically insulated (Emitting—E and Receiving—R) separated with a mortar sample (M) placed on a 150 × 150 mm^2^ gap. The perimeter of the sample was conveniently insulated to avoid acoustic bridges and edge noise. A 100 dBA pink noise sound source (A) was placed in the emitting compartment and the acoustic pressure was measured in thirds of an octave with three sound-level meters (S) placed inside both compartments and outside the acoustic chamber, according to the European standard UNE-EN ISO 10140-4 [33]).

A background noise of 50 dBA was recorded at the external sound-level meter and 40 dBA inside the closed acoustic chamber. Each mortar sample was measured every 15 s for 60 s on both sides: one corresponding to the side manufactured on the outer side of the mold, designated rough side (RG), and the other casted against the mold, denominated smooth side (SM). Acoustic pressure in dBA was measured for each frequency range between 100–5000 Hz, comparing measurements with and without mortar samples, as described in Figure 3. These experimental values were used to calculate the sound reduction index for each frequency range, and consequently calculate the global sound level inside the emitted (L_Ex_) and received (L_Rx_) compartments in dBA and the experimental values of critic frequency (ƒc) of the acoustic range (Figure 3).

The acoustic insulation was assessed according to the European standard UNE-EN ISO 717-1 [34] and Equations (1)–(3),
(1)$$R_{A}=L_{EX}-L_{RX}$$
(2)$$R_{A-C}=16.6\cdot \log m+5$$
(3)$$f_{c-c}=\frac{6.4\cdot 10^{4}}{d}\sqrt{\frac{D_{AP}\cdot \left(1-\nu^{2}\right)}{E}}$$
where R_A_ is the acoustic insulation or global sound reduction index in dBA of the mortar sample, R_A-C_ is the index estimated in dBA according to the acoustic mass law and f_c-c_ is the critic frequency in Hz calculated considering apparent density (D_AP_) in kg/m^3^, Young’s modulus € in N/mm^2^, Poisson’s ratio coefficient (ν), and sample thickness (d) in m.

## 3. Experimental Results: Acoustic Parameters Characterization and Analysis

Table 4 summarizes the experimental values of the acoustic parameters related to sound absorption (α_NRC_) and acoustic insulation (R_A-SM_, R_A-RG_, and R_A-C_). Mortars REF, P50, V50, V25, and P25CF30 showed low α_NRC_ values (0.037 ± 0.002). On the other hand, REFC, CF15, and A25 reached α_NRC_ values above 0.1. The different behavior can be related to the porous structure of lightweight aggregates and the net of voids among the gap-graded aggregate particles [6].

Acoustic insulation of the smooth side (R_A-SM_) varied between 20 and 25 dBA, while acoustic insulation measured on the rough surface of the sample was lower, ranging from 18–22 dBA, except REF (27.80 dBA) and V50 (25.70 dBA). V50 and P50 presented the largest R_A-SM_ values and CF15 the lowest. Regarding acoustic insulation of the rough side (R_A-RG_), REF showed the largest value, and P25CF30 was the lowest. Accordingly, it can be said that surface roughness influence acoustic insulation of lime-cement mortars in this study in most cases, except REF and V50, mixtures with larger content of fine particles that reduce surface roughness.

Acoustic insulation calculated according to Equation (2) (R_A-C_), considering the physical and mechanical properties of the mortars, was very similar for all the mixtures as expected, due to the larger influence of mass in Equation (2) and the slight differences of D_AP_. Calculated R_A-C_ values reached 31 ± 2 dBA, 4–15 dBA larger than those measured in the laboratory (R_A-SM_ and R_A-RG_).

The spectral frequency analysis showed the largest absorption values for 300 Hz and a decrease can be observed between 600 and 1000 Hz, followed by an increase until 1600 Hz. Acoustic reduction showed a maximum at 250 Hz and a second peak at 1600 Hz, with a critic frequency (*f*_c_) around 500 Hz. However, the critic frequency calculated according to Equation (3) (*f*_c-c_) would be between 860 and 1460 Hz, and the thirds of the octave that showed closer values were 800, 1000, and 1250 Hz.

## 4. Discussion: Acoustic Assessment of Multiscale Porous Lime-Cement Mortars

The effect of gap-graded aggregates (GGA), lightweight aggregates (LA), and fibers (F) on the acoustic properties of a multiscale porous lime-cement mortar (MP-LCM) is discussed. According to the literature [5], three groups of sound insulation parameters were considered: mass (D_AP_), homogeneity (P_O_ and P_V_), and stiffness (E, K, and ν). The acoustic properties of multiscale porous lime-cement mortars (MP-LCM) are then discussed. The relationship between airborne noise absorption (α_NRC_) and acoustic insulation (R_A_) is also analyzed. Finally, a model for the acoustic performance of MP-LCM is proposed.

### 4.1. Effect of MP-LCM Composition on Acoustic Properties

Gap-graded aggregate mortar (REFC) enhanced noise reduction coefficient (α_NRC_) up to 0.10, regarding mortar with continuously graded sand (REF). On the other hand, considering acoustic insulation of the smooth side (R_A-SM_), REFC reached a similar value to REF (23 dBA). However, acoustic insulation on the rough side of the REFC samples was 9 dBA lower than the REF sample.

When mixtures with lightweight aggregates (A25, P25, and V25) were considered, a change in acoustic properties regarding REFC was measured. Expanded clay (A25) slightly increased noise absorption up to 0.113, whereas perlite and vermiculite (P25 and V25) exhibited lower values than REFC. On the other hand, lightweight aggregates enhanced the acoustic insulation of the rough face. Only perlite (P25) showed better acoustic insulation performance of the smooth face, reaching the largest R_A-SM_ value (25 dBA). The proportion of LA (perlite and vermiculite) hardly influenced airborne noise absorption. Regarding acoustic insulation of MP-LCM, double LA volume slightly varied the R_A_ values, except for the rough face of V50. That is, vermiculite showed the maximum value of R_A-RG_ (25.70 dBA).

Cellulose fibers (CF) also modified the acoustic properties of MP-LCM. Sound absorption highly depended on the proportion of cellulose fibers, producing CF15 the maximum value of sound absorption (0.127). However, a larger number of fibers (CF30) did not enlarge the noise absorption coefficient (α_NRC_) of MP-LCM [8]. It was observed that cellulose fibers and the volumetric fraction of CF did not significantly change the acoustic insulation. CF15 did not improve REFC values, whereas CF30 only increased R_A-RG_ by 1 dBA. CF30 only achieved an extra 2 dBA of acoustic insulation regarding CF15.

The mortar mixture with perlite lightweight aggregate (P25) was combined with two types of short fibers: cellulose fibers (P25CF30) and polypropylene fibers (P25PPF). In these mixtures, the type of fibers modified the noise absorption differently: P25PPF increased the α_NRC_ value, whereas P25CF30 reduced it. On the other hand and regarding acoustic insulation, mortars with both types of fibers presented lower R_A_ values than P25, especially the rough side of P25CF30, which corresponded to the minimum R_A_ measured value.

### 4.2. Assessment of Airborne Sound Reduction

According to the acoustic mass law (Equation (2)), sound insulation can be related to superficial weight (R_A-C_). The calculated and the experimental acoustic insulation (R_A_) are compared in Figure 4. It can be observed that the laboratory measurements were lower than the calculated sound reduction indices because the superficial weight was very similar for all the multiscale porous lime-cement mortars (MP-LCM). Therefore, the effectiveness of MP-LCM to reduce noise transmission must also depend on other parameters [5]. The influence of mass (D_AP_), homogeneity (P_O_ and P_V_), and stiffness (E, K, and ν) on the acoustic insulation performance of MP-LCM was analyzed.

#### 4.2.1. Effect of Mass on Sound Insulation

Figure 5 plots apparent density (D_AP_) and experimental sound reduction index (R_A_) on the rough (RG) and smooth (SM) sides of the MP-LCM samples. The results showed an inverse relationship in a range of D_AP_ from 1300 to 2000 kg/m^3^. Thus, the larger the mass, the lower the acoustic insulation of multiscale porous lime-cement mortars. Apparent density was measured by filling with water the net of voids among the gap-graded aggregate particles and disregarding part of the volume of the sample. Therefore, changes in mass (D_AP_) on MP-LCM were not sufficient to achieve an improvement in acoustic insulation.

#### 4.2.2. Homogeneity and Sound Insulation Performance

According to experimental results, acoustic insulation (R_A_) and open porosity (P_O_) were directly related (Figure 6a) when the smooth and rough sides were compared. The use of GGA produced accessible voids, turning the mortars into pervious materials [11]. Large voids degrade the acoustic insulation performance because of sound diffraction [5,10].

However, the open porosity of MP-LCM increased due to paste fines content that filled the large voids of GGA and refined the mesopores network [6,8]. On the other hand, acoustic insulation and water vapor permeability (P_V_) showed an inverse relationship (Figure 6b). Water vapor permeability increased due to the increase in small pore size and a better-connected pore network. Consequently, it can be said that extra acoustic insulation was achieved by reducing water vapor permeability and increasing the open porosity of MP-LCM.

#### 4.2.3. Elastic Stiffness, Compressibility, and Airborne Noise Transmission

Some relations between sound insulation (R_A_) and mechanical parameters calculated from ultrasonic pulse velocity (E, K, and ν) were identified:The global sound reduction indices (R_A_) decrease when Young’s (E) and compressibility (K) moduli increased, in both types of surfaces (Figure 7). That is, larger stiffness and compressibility meant a reduction in the acoustic insulation performance of MP-LCM.

Airborne noise transmission (R_A_) and Poisson’s ratio (ν) were inversely proportional (Figure 8). Accordingly, it can be said that more compressible MP-LCM enhances acoustic insulation of the smooth side (R_A-SM_) and the rough side (R_A-RG_).

### 4.3. Acoustic Performance: Noise Reduction Coefficient and Sound Reduction Index

Figure 9 plots acoustic insulation (R_A_) against airborne noise absorption (α_NRC_). Larger noise absorption of MP-LCM was observed to slightly reduce sound insulation. Previously, selecting aggregate grading (GGA) enhanced the noise reduction coefficient due to a reduced open porosity and optimal void size [6,8]. Nevertheless, increasing open porosity improved the acoustic insulation of MP-LCM (Figure 6a). Accordingly, a compromise between noise insulation and absorption must be searched. This study demonstrated that MP-LCM can be designed to achieve acoustic insulation in a range from 18 to 24 dBA and a noise absorption above 0.10.

### 4.4. Multiscale Porous Lime-Cement Mortars: A Model for Acoustic Performance

Figure 10 presents a multiscale semi-quantitative model for acoustic performance, following the basic model for pervious lime-cement-based mortars described elsewhere [8]. At the macroscale, three phases were identified: a lime-cement shell (PS), spherical monotonic size aggregates (A_SMS_), and a continuous void network (CVN). Lime-cement shell thickness (d_PS_) and active void size (VAS) were associated, as active void size varied inversely to paste shell thickness [8]. At the paste shell microscale, a multiphase matrix was considered (cement gel, lime crystals, fines, fibers, and air). Paste thickness and type of microscale phases depended on the component type and volume of paste [6,8].

Sound absorption varied due to the relation between the active void size and the paste shell thickness [8]. Reducing the size of the voids meant decreasing the noise absorption coefficient (α_NRC_), as other authors have reported that creating open porosity in concrete improved sound absorption due to the internal friction within the void walls, airflow resistivity, and tortuosity [10,19].

The effect of lightweight aggregates (LA), such as perlite and vermiculite, on the paste shell thickness must also be discussed. LA increased the paste volume due to the filler effect of the large amount of LA dust produced by the fracture of LA particles [6] that thickened the shell and reduced the active void size, decreasing α_NRC_ [8]. Sound absorption barely changed when the amount of LA was doubled, as the void network was already clogged.

On the other hand, expanded clay absorbed more noise since its particles were not broken and therefore the volume of paste was lower [6] and the paste shell thinner. Cellulose fibers also thickened the paste shell, although the amount of fiber was affected differently. A greater amount of fiber may lead to agglomerations, which further increase the paste shell thickness [8]. In addition, when perlite and fibers were combined, different behaviors were observed depending on the fiber type and the volumetric fraction added. A larger number of cellulose fibers implied a thicker paste shell and lesser sized void, producing one of the lowest noise reduction coefficients.

According to the experimental results (Figure 9), the acoustic insulation performance of MP-LCM (R_A_) improved when sound absorption (α_NRC_) decreased. Therefore, sound insulation in MP-LCM will enhance when paste thickness increase. In addition, the paste shell is characterized by pore connectivity (CPP), fines, and/or fiber content. The pore connectivity of the paste is related to the water vapor permeability and some authors have correlated water vapor permeability, air permeability, and acoustic performance [9]. Perlite and vermiculite improved acoustic insulation as they reduced CPP. Although expanded clay (A) reduced CPP, it did not improve acoustic insulation due to the porous structure of the aggregate and larger particle size than perlite and vermiculite [6]. That means greater inertia to vibrate and less acoustic energy dissipation [10,12]. The different aggregate intra-particle pores affected the acoustical behavior [9]. Combining LA with fibers meant a lower acoustic insulation performance, as fibers increased pore connectivity, especially cellulose fibers.

## 5. Conclusions

This paper presents an experimental study to evaluate the effect of gap-graded aggregates (GGA), lightweight aggregates (LA), and fibers (F) on the acoustic performance of multiscale porous lime-cement mortars (MP-LCM). The experimental program comprised measuring physical and mechanical properties and assessing airborne noise absorption and acoustic insulation. Mass, homogeneity, and stiffness were correlated to the global index of airborne noise transmission. As a result, a semi-quantitative multiscale porosity model for the acoustic performance of lime-cement mortars with GGA, LA, and F was proposed. The main findings of the study are:The use of GGA increased airborne noise absorption. Improvements in sound insulation were obtained by replacing gap-graded natural aggregate with perlite, vermiculite, or expanded clay. Where a higher insulation performance was required, the smooth surface of the perlite sample (P25) or rough surface of vermiculite mortar (V50) were preferred. The use of expanded clay or a small number of cellulose fibers was a good way to enhance sound absorption. Adding cellulose or polypropylene fibers to perlite mortars did not improve acoustic insulation.The effectiveness of MP-LCM as an acoustic insulator depended not only on the acoustic mass law but also on the surface roughness and mortar mass, homogeneity, and stiffness. A larger acoustic insulation capacity was achieved using the casted against the mold side. Improvements in insulation were obtained by reducing the apparent density and ultrasonic Young’s and compressibility moduli. On the other hand, reducing porosity accessible to water worsened MP-LCM acoustic performance.The acoustic insulation performance of MP-LCM improved when airborne noise absorption was lower.The multiscale porous lime-cement mortar composition affected paste thickness, active void size, and connectivity of paste pores. These parameters were linked to airborne noise absorption and acoustic insulation. The thicker the paste-shell, the lesser the absorbed sound. An increase in the connectivity of paste pores reduced acoustic insulation performance.

## Figures and Tables

**Figure 1 materials-16-00322-f001:**
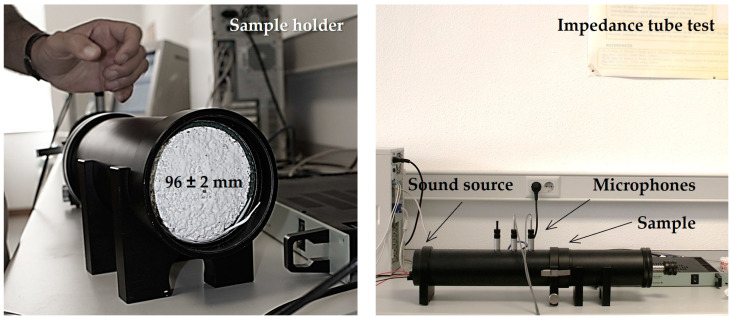
Impedance tube set-up for measuring sound absorption on cylindrical specimens.

**Figure 2 materials-16-00322-f002:**
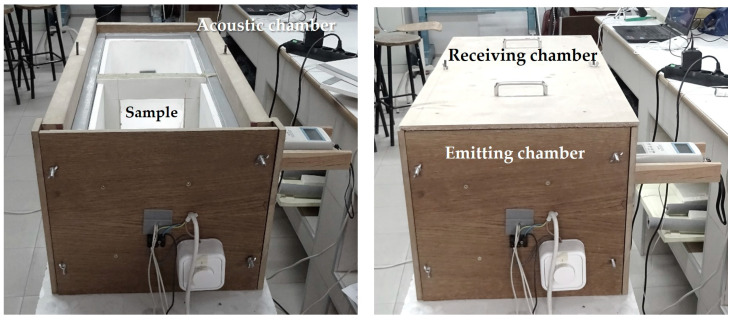
Acoustic reduced-scale chamber set-up for measuring sound insulation.

**Figure 3 materials-16-00322-f003:**
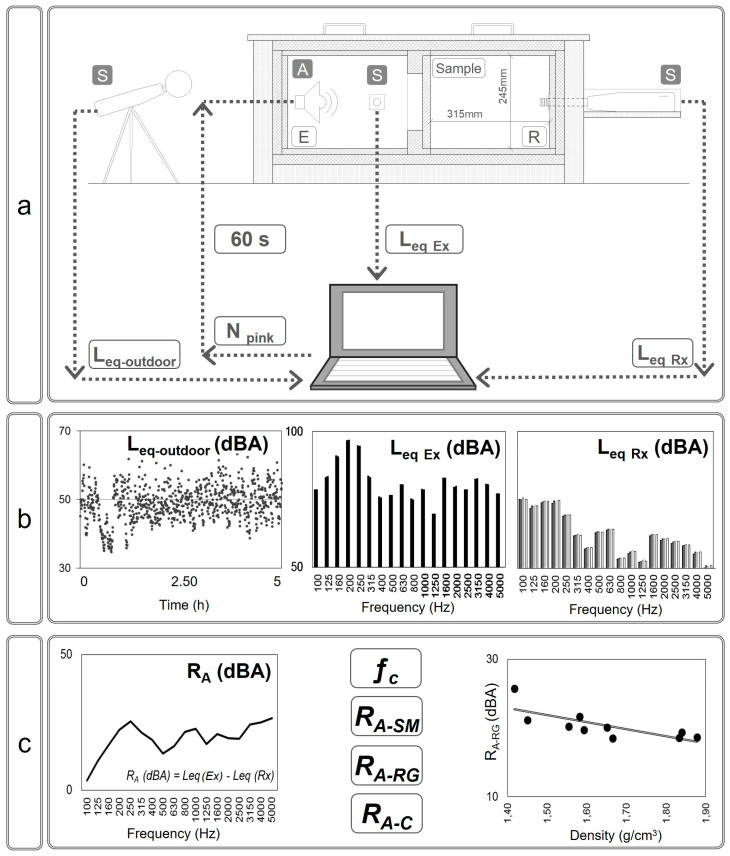
Scheme of measuring set-up and data analysis for sound insulation in multiscale porous lime-cement mortars: (**a**) Experimental work; (**b**) Data processing; (**c**) Assessment of acoustic performance.

**Figure 4 materials-16-00322-f004:**
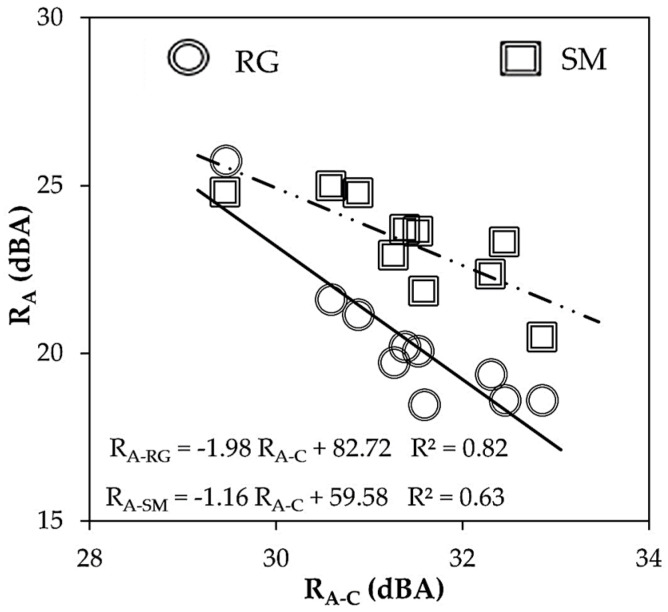
Experimental sound reduction index (R_A_) on rough (RG) and smooth (SM) sides versus estimated sound reduction index (R_A-C_), according to the acoustic mass law.

**Figure 5 materials-16-00322-f005:**
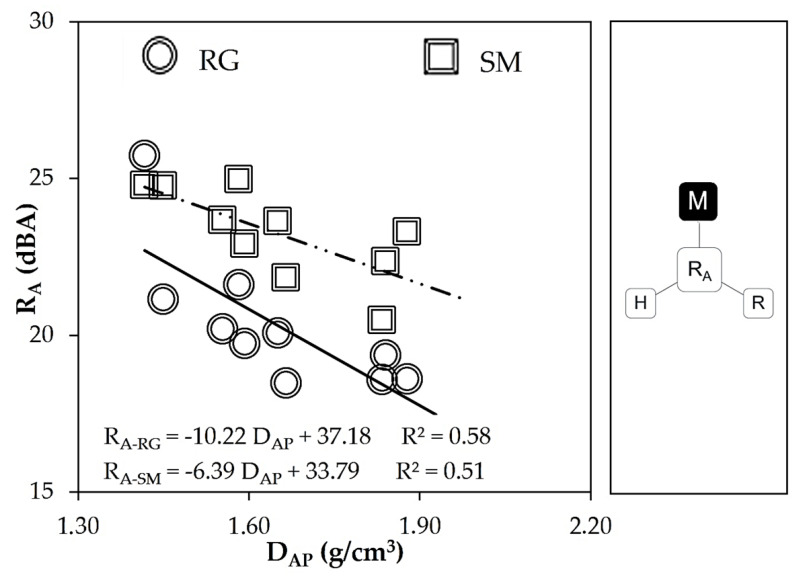
Quantification of the sound reduction index (R_A_) plotted against the weight per unit volume (D_AP_).

**Figure 6 materials-16-00322-f006:**
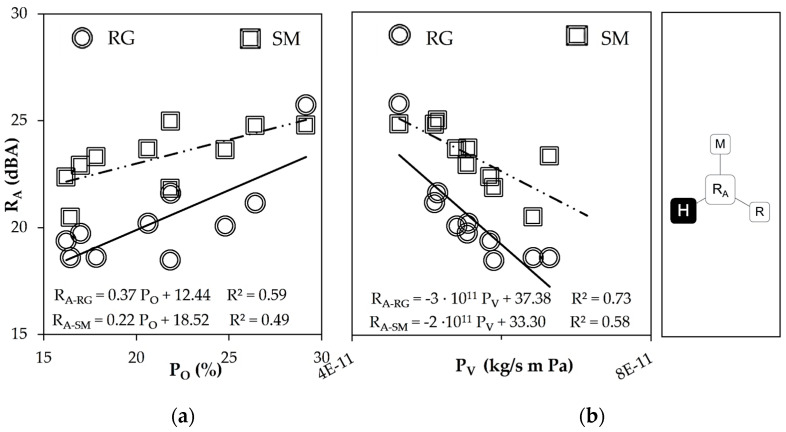
Porosity parameters related to sound reduction index (R_A_): (**a**) open porosity (P_O_) (**b**) and water vapor permeability (P_V_).

**Figure 7 materials-16-00322-f007:**
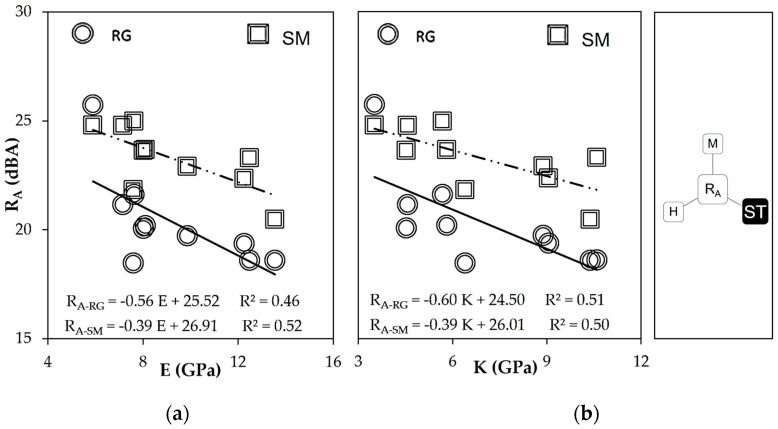
Mechanical parameters related to sound reduction index (R_A_): (**a**) Dynamic Young’s modulus (E) (**b**) and bulk modulus (K).

**Figure 8 materials-16-00322-f008:**
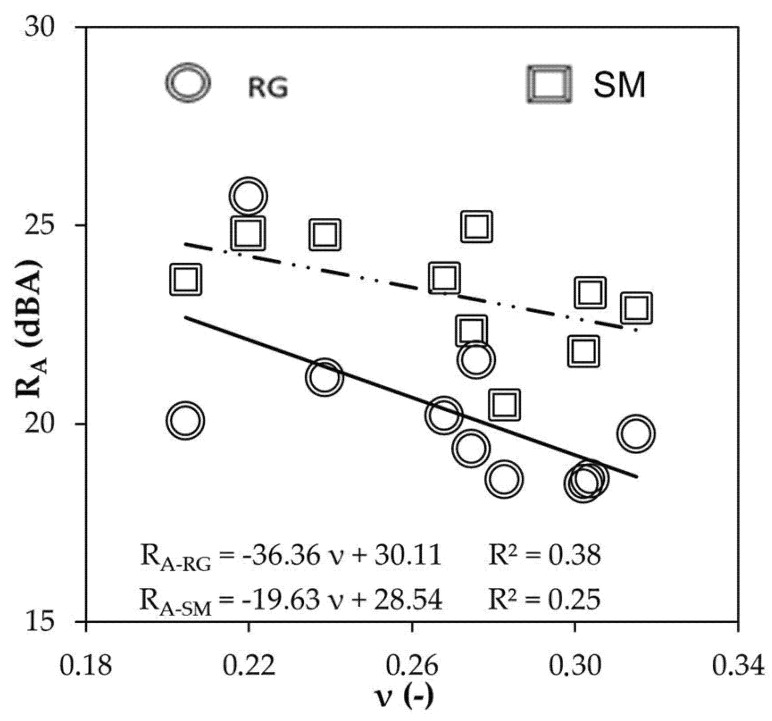
Experimental sound reduction index (R_A_) versus Poisson’s ratio (ν).

**Figure 9 materials-16-00322-f009:**
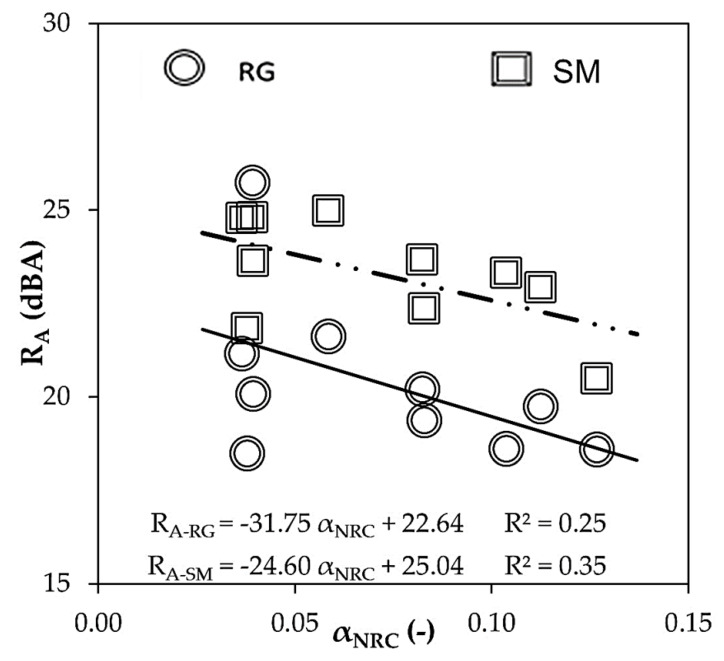
Experimental sound reduction index (R_A_) on rough (RG) and smooth (SM) sides versus noise reduction coefficient (α_NRC_).

**Figure 10 materials-16-00322-f010:**
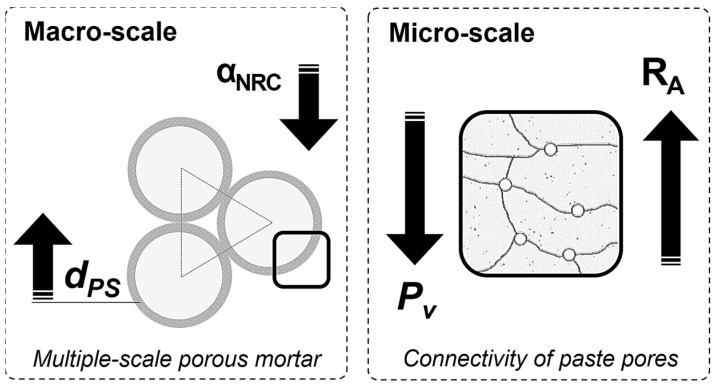
Semi-quantitative model for the acoustic performance of multiscale porous lime-cement mortars.

**Table 1 materials-16-00322-t001:** Components in kg for a batch of 1 m^3^ of multiscale porous lime-cement mortars (adapted from [6]).

Components	REF	REFC	CF15	CF30	A25	P50	P25	P25CF30	P25PPF	V50	V25
Cement	214	214	214	214	214	214	214	214	214	214	214
Lime	68	68	68	68	68	68	68	68	68	68	68
Cellulose fibers	-	-	0.8	1.6	-	-	-	1.6	-	-	-
PP fibers	-	-	-	-	-	-	-	-	0.81	-	-
Sand (0–4)	1379	-	-	-	-	-	-	-	-	-	-
Sand (2–3)	-	1502	1502	1502	1127	751	1127	1127	1127	751	1127
Expanded clay	-	-	-	-	82	-	-	-	-	-	-
Perlite	-	-	-	-	-	77	38	38	38	-	-
Vermiculite	-	-	-	-	-	-	-	-	-	82	41
Water ^1^	200	140	140	140	160	260	220	220	225	300	270
w/b ^2^	1.08	0.56	0.56	0.56	0.61	0.96	0.83	0.83	0.85	1.10	1.01

^1^ Liquid water added. ^2^ LA absorption water and sand humidity were also taken into account.

**Table 2 materials-16-00322-t002:** Physical properties related to mortar microstructure (adapted from [6,30]).

Parameters	REF	REFC	CF15	CF30	A25	P50	P25	P25CF30	P25PPF	V50	V25
D_AP_ (g/cm^3^) ^1^	1.81	1.88	1.83	1.84	1.59	1.45	1.58	1.67	1.55	1.42	1.65
P_O_ (%)	25.62	17.83	16.42	16.22	17.01	26.41	21.85	21.83	20.64	29.16	24.79
C (kg/m^2^ min^0.5^)	1.30	0.53	0.65	0.55	0.50	1.08	0.88	1.07	0.83	1.30	1.48
P_v_ 10^−11^ (kg/m s Pa)	5.13	6.64	6.43	5.85	5.54	5.10	5.14	5.90	5.56	4.63	5.40

^1^ Apparent density was calculated using a hydrostatic balance.

**Table 3 materials-16-00322-t003:** Mechanical properties of multiscale porous lime-cement mortars by ultrasonic pulse transmission technique (adapted from [30]).

Parameters	REF	REFC	CF15	CF30	A25	P50	P25	P25CF30	P25PPF	V50	V25
E (GPa)	9.18	12.49	13.55	12.25	9.87	7.16	7.64	7.60	8.10	5.90	8.02
K (GPa)	5.28	10.60	10.39	9.06	8.89	4.57	5.68	6.40	5.82	3.51	4.53
ν (-)	0.21	0.30	0.28	0.27	0.31	0.24	0.28	0.30	0.27	0.22	0.20

**Table 4 materials-16-00322-t004:** Acoustic properties of multiscale porous lime-cement mortars (adapted from [6]).

Parameters	REF	REFC	CF15	CF30	A25	P50	P25	P25CF30	P25PPF	V50	V25
α_NRC_ (-)	0.035	0.104	0.127	0.083	0.113	0.037	0.059	0.038	0.083	0.039	0.039
R_A-SM_ (dBA)	23.40	23.30	20.50	22.40	22.90	24.80	25.00	21.80	23.70	24.80	23.60
R_A-RG_ (dBA)	27.80	18.60	18.60	19.40	19.70	21.20	21.60	18.50	20.20	25.70	20.10
R_A-C_ (dBA)	32.20	32.50	32.90	32.30	31.30	30.90	30.60	31.60	31.40	29.50	31.50
f_c-c_ (Hz)	1156	985	869	994	1017	1119	1273	1190	1080	1457	1184

## Data Availability

Authors can confirm that all relevant data are included in the article.

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
