# Peer review of "Acoustic Assessment of Multiscale Porous Lime-Cement Mortars"

_materials, 2022, doi:10.3390/ma16010322_

Round 1

Reviewer 1 Report

In general, the subject treated in the article "Acoustic Assessment of Multiscale Porous Lime-Cement Mortars" by I. Palomar and G. Barluenga is very interesting. The article shows novel results in the field of acoustic measurements in porous mortars. However, the article needs minor changes to be published in Materials.

The following is a list of observations:

 1. Abstract: It is very important for readers to know the percentages used for each component (lines 12-14), as well as the percentages of improvement, in this case the sound insulation (lines 21-22).

2. Keywords: add the terms: polymer fibers, expanded clay, perlite, vermiculite.

3. In the Introduction section, add some references to the use of polypropylene as a sound reducer in mortars.

4. Line 76, maybe the abbreviation for cellulose fibers is (CF) instead of (FC), as well as (PPF) for polypropylene fibers instead of (FPP).

5. For the readers it is difficult to follow the results shown in Tables 2 and 3, it is better to show the results in graphs.

Author Response

Ms. Ref. No.: materials- 2084309

Title: Acoustic Assessment of Multiscale Porous Lime-Cement Mortars.

Materials: Special Issue: Acoustic Properties of Materials

Response to Reviewer 1 comments (round 1):

  1. Abstract: It is very important for readers to know the percentages used for each component (lines 12-14), as well as the percentages of improvement, in this case the sound insulation (lines 21-22).

As suggested by the reviewer, some information has been included:

“Gap-graded sand was replaced by 25 and 50 % of lightweight aggregates. 1.5% and 3% of cellulose fibers by volume were added.”

“The incorporation of lightweight aggregates increased up to 38% sound insulation compared to the gap-graded sand reference mixture. Fibers slightly improved sound insulation, although small fraction of cellulose fibers can quadruplicate noise absorption.”

  1. Keywords: add the terms: polymer fibers, expanded clay, perlite, vermiculite.

Keywords have been added.

  1. In the Introduction section, add some references to the use of polypropylene as a sound reducer in mortars.

Some references have been added to complete the introduction, according to the reviewer’s suggestion:

[19] Amran, M.; Fediuk, R.; Murali, G.; Vatin, N.; Al-Fakih, A. Sound-Absorbing Acoustic Concretes: A Review. Sustainability 202113, 10712. https://doi.org/10.3390/su131910712.

[21] Alyousef, R.; Mohammadhosseini, H.; Ebid, A.A.K.; Alabduljabbar, H.; Ngian, S.P.; Huseien, G.F.; Mohamed, A.M. Enhanced Acoustic Properties of a Novel Prepacked Aggregates Concrete Reinforced with Waste Polypropylene Fibers. Mater. 2022, 15(3), 1173. https://doi.org/10.3390/ma15031173.

  1. Line 76, maybe the abbreviation for cellulose fibers is (CF) instead of (FC), as well as (PPF) for polypropylene fibers instead of (FPP).

Abbreviation have been revised as suggested.

  1. For the readers it is difficult to follow the results shown in Tables 2 and 3, it is better to show the results in graphs.

The authors consider that the presented tables are adequate for the purpose of this study. Their aim is to give an overview of physical and mechanical properties as these results have been previously published and have been adapted.

Reviewer 2 Report

This paper reports the acoustic performance of eleven multiscale porous lime-cement mortars (MP-LCM) with two types of fibers (cellulose and polypropylene), a gap-graded sand and three lightweight aggregates (expanded clay, perlite, and vermiculite). It's well organized and structured. There are some issues needed to be clarified before it can be accepted for publication:

1.       Pls delete these unnecessary and unfamiliar abbreviations from the paper. The authors should explain or annotate the abbreviations at their first use site. It is beneficial to readers who are not familiar with this field. E.g. lightweight aggregates (LA): expanded clay (A), perlite (P) and ver- 74

2.       The manuscript should be very carefully checked to avoid any errors. The language should be checked throughout the text and any grammar mistakes should be corrected.

3.       There are many studies on lightweight concrete or sound-absorbing concrete, and the authors should discuss lightweight concrete in the Introduction. E.g. Insulating and fire-resistance performance of calcium aluminate cement based lightweight mortars; A green ultra-lightweight chemically foamed concrete for building exterior: A feasibility study.

4.       The theoretical analysis needs a more in-depth discussion.

5.       At the end of the introduction, it is necessary to clearly identify the goal and tasks for achieving it, and in the conclusions, give a numbered list of tasks solved.

6.       The research contributions of the paper should be articulated more clearly. The abstract is not representative of the content and contributions of the paper. The abstract does not seem to properly convey the rigor of research.

7.       Conclusions should be simplified.

Author Response

Response to Reviewer 2 comments (round 1):

  1. Pls delete these unnecessary and unfamiliar abbreviations from the paper. The authors should explain or annotate the abbreviations at their first use site. It is beneficial to readers who are not familiar with this field. E.g. lightweight aggregates (LA): expanded clay (A), perlite (P) and ver- 74.

The abbreviation for the three types of lightweight aggregates has been included when the components were described (page 2). These abbreviations may be useful for readers as they also refer to the mixture labels (Table 1). On the other hand, some abbreviations from Conclusions have been erased to simplify the text.

  1. The manuscript should be very carefully checked to avoid any errors. The language should be checked throughout the text and any grammar mistakes should be corrected.

The manuscript has been checked as suggested.

  1. There are many studies on lightweight concrete or sound-absorbing concrete, and the authors should discuss lightweight concrete in the Introduction. E.g. Insulating and fire-resistance performance of calcium aluminate cement based lightweight mortars; A green ultra-lightweight chemically foamed concrete for building exterior: A feasibility study.

Regarding to lighten up cement-based materials and enhance acoustic properties, authors had been included some references as: Branco 2013, Rashad 2016, Pokorný 2022 and D’Alessandro 2018. In addition, some references have been added to complete the introduction, according to the reviewer’s suggestion:

* Foam agent:

[10] Fediuk, R.; Amran, M.; Vatin, N.; Vasilev, Y.; Lesovik V.; Ozbakkaloglu, T. Acoustic Properties of Innovative Concretes: A Review. Mater. 2021, 14(2), 398. https://doi.org/10.3390/ma14020398.

* Polymer fibers:

[19] Amran, M.; Fediuk, R.; Murali, G.; Vatin, N.; Al-Fakih, A. Sound-Absorbing Acoustic Concretes: A Review. Sustainability 202113, 10712. https://doi.org/10.3390/su131910712.

[21] Alyousef, R.; Mohammadhosseini, H.; Ebid, A.A.K.; Alabduljabbar, H.; Ngian, S.P.; Huseien, G.F.; Mohamed, A.M. Enhanced Acoustic Properties of a Novel Prepacked Aggregates Concrete Reinforced with Waste Polypropylene Fibers. Mater. 2022, 15(3), 1173. https://doi.org/10.3390/ma15031173.

  1. The theoretical analysis needs a more in-depth discussion.

Some extra explanations have been included in section 4, Discussion: Acoustic Assessment of Multiscale Porous Lime-Cement Mortars, as suggested by the reviewer:

“Large voids degrade the acoustic insulation performance because of sound diffraction [5, 10].”

“Reducing the size of the voids meant decreasing noise absorption coefficient (αNRC), as other authors have reported that creating open porosity in concrete improved sound absorption due to the internal friction within the void walls, airflow resistivity and tortuosity [10, 19].”

“Although expanded clay (A) reduced CPP, it did not improve acoustic insulation due to the porous structure of the aggregate and larger particle size than perlite and vermiculite [6]. That means greater inertia to vibrate and less acoustic energy dissipation [10, 12].”

  1. At the end of the introduction, it is necessary to clearly identify the goal and tasks for achieving it, and in the conclusions, give a numbered list of tasks solved.

The goal and tasks have been identified and numbered at the end of the Introduction:

“The present study aims to evaluate the acoustic performance of lime-cement mortars regarding airborne noise using reduced scale tests. The tasks for achieving this goal are:

  1. Discussing the effect of gap-graded aggregates, lightweight aggregates and fibers on airborne noise absorption and acoustic insulation of multiscale porous lime-cement mortars.
  2. Considering the influence of physical and mechanical parameters over sound insulation performance of multiscale porous lime-cement mortars.
  3. Analyzing the relationship between sound absorption and sound insulation parameters of multiscale porous lime-cement mortars.
  4. Describing a multiscale semi-quantitative model for acoustic performance of porous lime-cement mortars.”
  5. The research contributions of the paper should be articulated more clearly. The abstract is not representative of the content and contributions of the paper. The abstract does not seem to properly convey the rigor of research.

As suggested by the reviewer, abstract has been improved:

“Gap-graded sand was replaced by 25 and 50 % of lightweight aggregates. 1.5% and 3% of cellulose fibers by volume were added.”

“The incorporation of lightweight aggregates increased up to 38% sound insulation compared to the gap-graded sand reference mixture. Fibers slightly improved sound insulation, although small fraction of cellulose fibers can quadruplicate noise absorption.”

  1. Conclusions should be simplified.

As suggested by the reviewer, Conclusions have been improved. The findings have been simplified and numbered. These numbers correspond to the goal and tasks from the Introduction:

The main findings of the study are:

  1. The use of GGA increased airborne noise absorption. Improvements in sound insulation were obtained by replacing gap-graded natural aggregate with perlite, vermiculite or expanded clay. Where a higher insulation performance was re-quired, smooth surface of perlite sample (P25) or rough surface of vermiculite mortar (V50) were preferred. The use of expanded clay or a small amount of cellulose fibers was a good way to enhance sound absorption. Adding cellulose or polypropylene fibers to perlite mortars did not improve acoustic insulation.
  2. The effectiveness of MP-LCM as an acoustic insulator depended not only on the acoustic mass law but also on the surface roughness and mortar mass, homogeneity, and stiffness. Larger acoustic insulation capacity was achieved using the casted against the mould side. Improvements in insulation were obtained by reducing the apparent density and ultrasonic Young and Compressibility Moduli. On the other hand, reducing porosity accessible to water worsened MP-LCM acoustic performance.
  3. Acoustic insulation performance of MP-LCM improved when airborne noise absorption was lower.
  4. The multiscale porous lime-cement mortar composition affected paste thickness, active void size and connectivity of paste pores. These parameters were linked to the airborne noise absorption and acoustic insulation. The thicker the paste-shell, the lesser the absorbed sound. An increase of connectivity of paste pores reduced acoustic-insulation performance.”

Round 2

Reviewer 2 Report

The authors have revised the manuscript.